# Nanogenerator-based dual-functional and self-powered thin patch loudspeaker or microphone for flexible electronics

Wei Li[1], David Torres[2], Ramón Díaz[2], Zhengjun Wang[3], Changsheng Wu[3], Chuan Wang[2], Zhong Lin Wang[3] & Nelson Sepúlveda[1,2]

Ferroelectret nanogenerators were recently introduced as a promising alternative technology for harvesting kinetic energy. Here we report the device's intrinsic properties that allow for the bidirectional conversion of energy between electrical and mechanical domains; thus extending its potential use in wearable electronics beyond the power generation realm. This electromechanical coupling, combined with their flexibility and thin film-like form, bestows dual-functional transducing capabilities to the device that are used in this work to demonstrate its use as a thin, wearable and self-powered loudspeaker or microphone patch. To determine the device's performance and applicability, sound pressure level is characterized in both space and frequency domains for three different configurations. The confirmed device's high performance is further validated through its integration in three different systems: a music-playing flag, a sound recording film and a flexible microphone for security applications.

[1] Department of Mechanical Engineering, Michigan State University, East Lansing, Michigan 48824, USA. [2] Department of Electrical and Computer Engineering, Michigan State University, East Lansing, Michigan 48824, USA. [3] School of Materials Science and Engineering, Georgia Institute of Technology, Atlanta, Georgia 30332-0245, USA. Correspondence and requests for materials should be addressed to N.S. (email: nelsons@egr.msu.edu).

The interest in ultrathin, flexible paper-like personal electronics is growing at an accelerated pace, and the pursuit of multifunctionality and wearability has become a major technological trend[1–12]. The continuous, aggressive and multidisciplinary research efforts from multiple groups has created a sustained stream of innovation that has been constantly evolving and improving; making a large number of innovative products such as wearable computer screens, electronic newspapers, flexible light-emitting diodes, artificial skin and smart gloves within the foreseeable future[13–21]. User–device interaction is a very important aspect of wearable technologies. It is understandable that from the five recognized human senses (that is, vision, audition, gustation, olfaction and somatosensation[22]), most of the current advances are focused on the use of sight and touch. However, the optimal wearable system designs should consider all human senses, since this will enable all the possible user–device interactions and allow for the use of the most appropriate one, depending on the application. Besides vision and touch, acoustic interaction (that is, hearing and speaking) also has good potential of becoming an essential and convenient bridge between human and flexible personal electronics in the coming years. In a way, we are experiencing this already through the voice-activated gadgets, speech-to-text converters and audio recognition systems.

The prevailing loudspeakers used today consist at least of a diaphragm, a voice coil attached to the apex of the diaphragm, a permanent magnet fixed to the loudspeaker's frame via a flexible suspension and an enclosure[23]. When an audio current waveform is applied to the voice coil, an audio frequency movement of the diaphragm is produced due to the magnetic interaction between the voice coil and the magnet, thereby reproducing the sound pressure waves[24]. Recently, a thermoacoustic superaligned carbon nanotube loudspeaker was demonstrated that uses thermal expansion and contraction of the medium to generate sound waves[23,25–27]. A microphone is a reversely operating device, in which sound waves are converted to electrical signals, functions and it can be considered as the 'ears' of flexible personal devices. Recently, the use of a thin, rollable, triboelectric nanogenerator was demonstrated to harvest sound waves[28]. The further advance of human–computer interaction in wearable electronics would benefit from thin-film, flexible, light-weight and robust devices that can serve as both: loudspeaker and microphone.

In this communication, a dual-functional and self-powered thin-film flexible acoustic transducer that operates as both loudspeaker and microphone is reported. Based on ferroelectret nanogenerator (FENG) reported recently[29], the device has the ability to produce power as well. By means of microplasma discharging, the artificial voids inside the foam-structured FENG forms numerous giant dipoles that enable the FENG with outstanding electromechanical transformation efficiency. We describe the mechanisms for direct and reverse interaction effects (that is, using mechanical energy to produce electrical energy, and using electrical energy to produce mechanical energy) that are the cornerstones of the present FENG-based loudspeaker/microphone device. Characterization of the performance of the FENG-based loudspeaker is done by sound pressure level (SPL) directivity measurements for three developed configurations: free-standing FENG, FENG attached to a soft substrate and rolled FENG in cylindrical shape. Moreover, the amplitude-frequency response (10–20,000 Hz) for the present FENG-based loudspeaker is characterized as a function of functional area, number of layers and for cases with or without substrate. To demonstrate the applicability and functionality of the device, we fabricated a FENG-based music-playing flag that can operate under regular wind conditions. Due to the electromechanical coupling of the device, the operation of the FENG-based loudspeaker can be reversed, to result in a FENG-based microphone that converts acoustic vibration (sound) into an electrical signal. The FENG-based microphone is highly sensitive to a broad range of frequency. This is shown by the other two demonstrations presented in this work; these include the recording of a symphony by a piece of paper-like FENG-based microphone with high fidelity, and the security of a personal computer by the voiceprint identity recognition through a FENG-based device that can easily be integrated secretly within a computer or personal device for added subtlety.

## Results

**Structural configuration.** The schematic structure of the flexible FENG-based acoustic transducer is shown in Fig. 1a. The total thickness of the device is < 100 µm. The functional material of the FENG-based acoustic transducers is prepared by starting with polypropylene film containing tiny foreign silicates particles (0.1–10 µm), as shown in Supplementary Fig. 1. Polypropylene (PP) is a type of commodity plastic with the merits of low density, high flexibility and good resistance to fatigue[30,31]. Once the PP film experiences stretching in two perpendicular directions, the inorganic particles serve as stress concentrators or micocracks, allowing the film to be filled with lens-shaped voids with diameters ranging from 1 to 100 µm. During this process, high pressure (for example, 5 MPa) nitrogen or carbon dioxide gas is diffused into the PP film, so that the internal pressure within the voids becomes equal to the external pressure. Next, the external gas pressure is suddenly released, resulting in dramatically swell of those voids in PP film. For the purpose of stabilizing and stiffening the swelling voids at room temperature, thermal treatment (usually > 100 °C) is carried out to increase the crystallinity of the polymer matrix[32–34]. Subsequently, by applying a large electric field to the treated film, Paschen breakdown occurs inside the voids. The current within the air gap transfers a sheet charge density across the air gap. During microplasma discharge, charges separated by the ionization of the

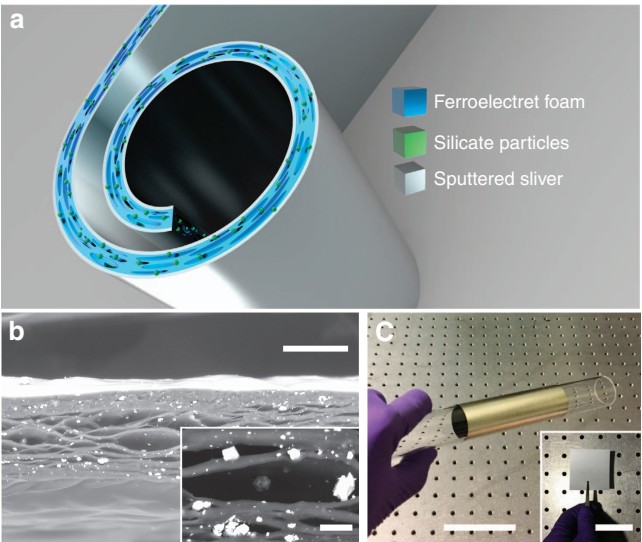

**Figure 1 | Schematic and experimental structure of FENG-based acoustic device.** (**a**) Schematic structure of a large area flexible FENG-based acoustic device. (**b**) Cross-sectional backscattered SEM image of the PP foam (scale bar, 20 µm). The inset shows the expanded view (scale bar, 10 µm), where the silicate particles can more easily be identified as the brighter regions. (**c**) Optical images of FENG-based acoustic device along the axial of a glass tube (scale bar, 5 cm). The inset shows the top view of PP foam inset (scale bar, 5 cm).

gas transportation under the charging field and light flashes can be observed with the naked eye, forming a 'microstorm'. After microplasma discharging, two thin layers of silver (500 nm) were deposited on both top and bottom side of PP film (80 μm) by sputtering. Scanning electron microscopyimages of the cross-section of PP foam are shown in Fig. 1b. Herein, backscattered electrons are used for revealing good contrast and clear definition of the structure. As the production of backscattered electrons is strongly dependent on the average atomic number of the sample, silicate particles appeared to be much brighter than their surrounding structure. Figure 1c shows the optical images of FENG-based acoustic device rolled along the axial of a glass tube. The highly flexible fabricated device consists of a stacked metal-insulator-metal thin-film structure without moving parts or microfabrication features or suspended structures, making the process easily scalable to large-scale fabrication.

**Working mechanism for reversible operation.** By means of microplasma discharges in the fabricated foam structure in PP film, opposite charges accumulate on the upper and lower surfaces of the artificial voids that in turn form numerous highly oriented giant dipoles, as shown in Fig. 2a. Once two layers of silver are sputtered on both outer surfaces of PP film, the giant dipoles in the PP film induce charges of opposite polarity in each silver film. Unlike piezoelectric material that have spontaneous electric polarization, micro-discharging treatment turns the PP film from completely nonpolar materials into artificial intelligent material that mimics both the microscopic molecular structure and macroscopic electromechanical behaviour of the piezoelectric material. Furthermore, in comparison with traditional piezoelectric materials, FENG features with flexibility and internally charged cellular structures, making them highly efficient in charge storage and more sensitive to mechanical stress. The direct electromechanical interaction effect of FENG is illustrated in Fig. 2b,c, and detailed information is shown in Supplementary Figs 2 and 3. When the FENG experiences compression or expansion in the thickness direction, the internal dipole moments simultaneously change in magnitude according to the applied pressure. Consequently, the change of dipole moments drive the compensation electrons from the electrode with negative charge to the electrode with positive charge, creating a potential between the two electrodes under open-circuit condition, or a flow of charges under closed-circuit condition. The electrical output of FENG was characterized and shown in Supplementary Fig. 4. On the reverse case, if extra charges are transferred to surface electrodes (or if there is a potential difference between them), the change of the charge density on the surface electrodes (or the electric field field across the thickness) reshape the giant dipoles inside the FENG, showing the reverse electromechanical interaction effect (Fig. 2d,e). Depending on the extra charge or magnitude of the applied voltage, numerous giant dipoles contract or expand at the same time, resulting in a change in the thickness of FENG. If the added charge or electric field is varied as a function of time, the changes in thickness of the FENG will also occur as a function of time, producing vibrations of its surfaces. This electromechanical coupling endows the FENGs with the capability of efficiently converting energy between the electrical and mechanical domains. This bidirectional capacity of the FENG adds to the list of device attributes (for example, simplicity, flexibility, softness, durability, lightweight, easy scaled-up structure) that not only broadens the device applicability from energy harvesting devices[29] to electroacoustic flexible devices, but also establishes a new operating principle for flexible loudspeakers and microphones.

**SPL directivity of FENG-based thin-film loudspeaker.** SPL directivity is a quantitative measure of the focusing of acoustic or sound energy. To investigate the frequency response change of FENG-based loudspeaker at off-axis angles, SPL directivity measurements were performed for three types of configurations: free-standing flat structure, flat structure with a transparent soft layer as the substrate and cylindrical structure. Since the proposed paper-like FENG-based thin-film loudspeaker is able to produce sound levels comparable to those produced by bulky commercial voice-coil loudspeakers, the acoustic performance can be directly tested by using the same measurement equipment for conventional acoustic products. The measurement setup for the SPL directivity is shown in Fig. 3a. The configuration shown in the setup is the free-standing FENG-based loudspeaker, but the same setup was used for the other two configurations. A piece of circular free-standing FENG-based loudspeaker (diameter of 8 cm) is clamped in the centre of a frame. A circular shape is chosen due to its central symmetry that aims at avoiding boundary effect during the film's movement. In the measurement, a prepolarized microphone system pointing to the centre of the FENG-based

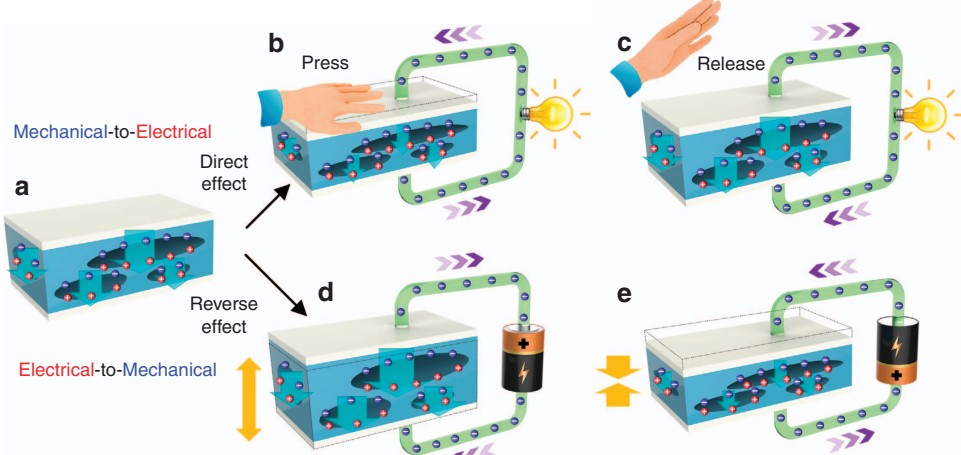

**Figure 2 | Energy conversion mechanisms of FENG.** (**a**) Charge distribution and giant dipoles of FENG after microplasma discharging, showing that the upper and lower surfaces of voids are oppositely charged. (**b,c**) Direct electromechanical interaction effect. (**b**) Pressed by human hand on the surface of FENG. (**c**) Pressure released and giant dipoles restore original sizes. (**d,e**) Reverse electromechanical interaction effect. (**d**) Giant dipoles further expand as positive potential is applied. (**e**) Giant dipoles shrink as negative potential is applied.

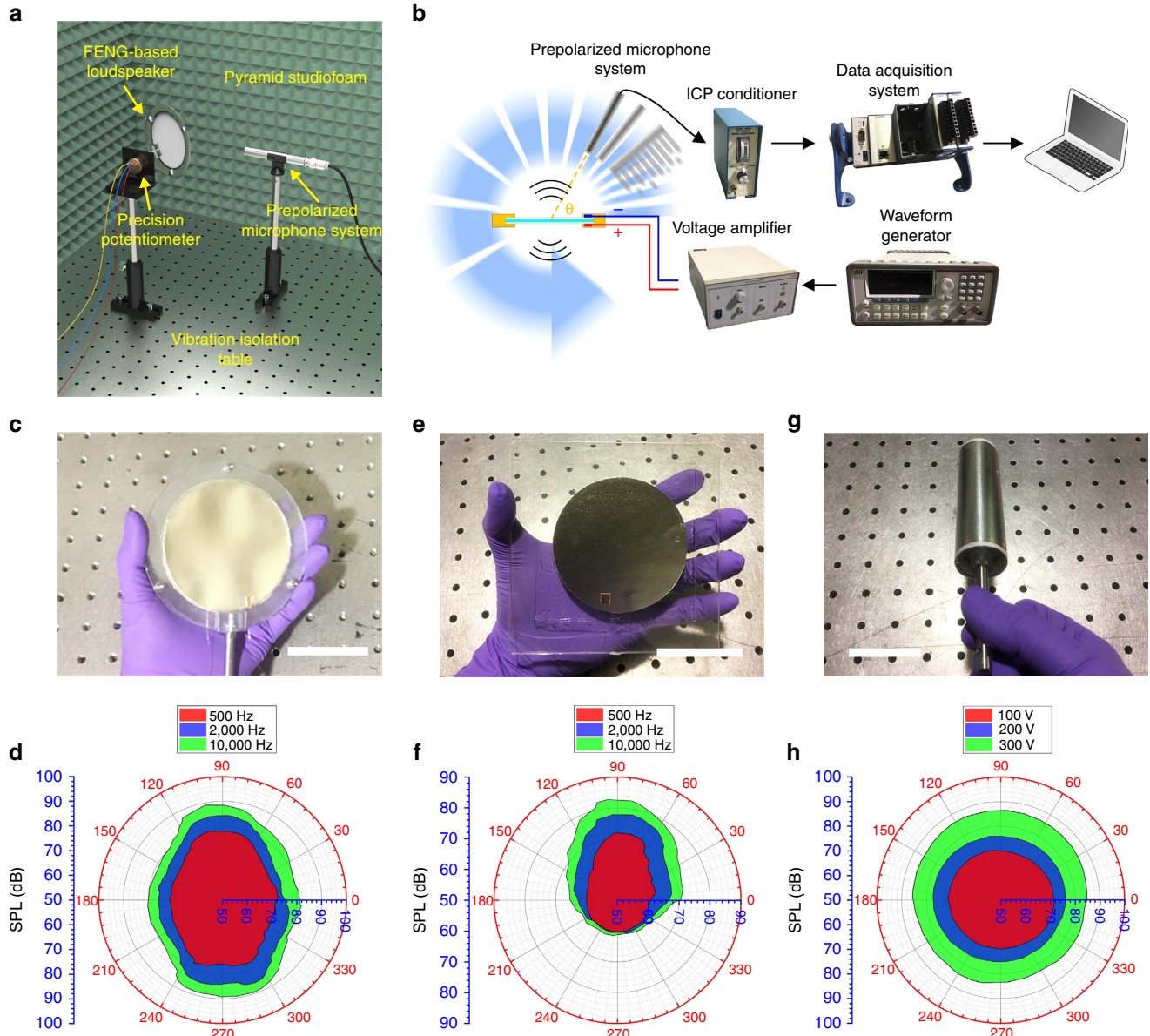

**Figure 3 | SPL directivity of three types of FENG-based loudspeaker.** (**a**) Experimental setup for free-standing FENG-based loudspeaker. (**b**) Lateral view illustration of measurement process with hardware connections. For the experiments, the FENG loudspeaker remained static, while the microphones was rotated. (**c**–**h**) Optical images and SPL polar plots under different frequency or voltage (scale bar, 5 cm). (**c**,**d**) Circular free-standing type. (**e**,**f**) Circular type with soft acrylic as substrate. (**g**,**h**) Rolled cylindrical type.

loudspeaker was rotated continuously and smoothly around the device. During rotation, the corresponding angle between the microphone and a vector normal to the device was recorded by precision potentiometer synchronously. In principle, keeping the device stationary while rotating the microphone will result in the same sound pattern measurement than keeping the microphone stationary, while rotating the device. Figure 3b illustrates the equivalent SPL directivity measurement process, the hardware used and their connections. The FENG-based loudspeaker was considered as a circular radiator, while the microphone measured SPL along a complete ring (that is, 360°) of uniform radius centered at the device that is also the origin of the polar plot. SPL directivity measurements indicate how much sound is directed towards a specific area as compared with all the sound energy being generated by a source. Polar plots of SPL directivity of circular free-standing FENG-based loudspeaker (Fig. 3c) with amplitude of 300 V and frequencies of 500 Hz, 2 kHz and 10 kHz

are shown in Fig. 3d. It can be noticed that the directional pattern of flat paper-like FENG loudspeaker is symmetric along the horizontal line, indicating that it can produce identical sound pressure from both surfaces. In comparison with most directional loudspeaker in the market, radiating sound from not only the frontside but also the backside at same level could allow more sound to be directed to more listeners in practical applications. The SPL for the free-standing configuration is related to the relative position between the device and the microphone. Overall, a larger projected area results in more sound wave radiated into the microphone and a larger SPL. In addition, the SPL directivity under 500 Hz, 2 kHz and 10 kHz shares similar directivity patterns except for their envelope sizes that is due to the amplitude-frequency characteristics. Figure 3e,f show a circular FENG-based loudspeaker (diameter of 8 cm) firmly adhered to a soft acrylic layer and the SPL directivity, respectively. The acrylic layer blocks sound radiation produced from the attached device's surface,

increasing the directivity of the FENG loudspeaker. For frequencies of 500 Hz, 2 kHz and 10 kHz, the SPL acquired from the backside are at the same level, whereas the SPL acquired from the frontside shows higher levels than the backside and with higher envelopes for higher frequencies. Compared with the free-standing configuration, the one-surface-blocked FENG-based loudspeaker attenuates great part of the radiation emitted in the substrate's direction; however, it has the added advantages of robustness and ease for installation that could be found very beneficial for industry and civil applications. The third FENG loudspeaker configuration consists of a rolled-up device in a cylindrical shape (Fig. 3g) with both perimeter and length of 8 cm. The SPL directivity 5 kHz for different input voltages (100, 200 and 300 V) is shown in Fig. 3h. The central symmetry makes this device an omnidirectional loudspeaker that theoretically would have the same directivity factor at every listener angle. Moreover, the SPL increases with the input voltage because larger voltage would increase the amplitudes of expansion or contraction of giant dipoles inside the FENG, so as to produce larger displacement in thickness direction and create larger sound pressure.

**Amplitude-frequency response and music-playing flag**. Further characterization is carried out by measuring SPL versus frequency response from 10 Hz to 20 kHz (Supplementary Fig. 5). In this case, the positions of both FENG-based loudspeakers and the microphone system are fixed. A network/spectrum analyser is used for sweeping the frequency of an voltage signal applied to the FENG-based loudspeaker through a low-noise voltage amplifier and also to measure the output voltage of the microphone system. The curve displayed in the network analyser represents the SPL spectrum. The first FENG-based loudspeaker configuration consists of squared free-standing configurations with different areas: 4 cm × 4 cm, 5.7 cm × 5.7 cm and 8 cm × 8 cm. Figure 4a shows the SPL frequency spectrum for the three devices. It can be seen that the amplitude of SPL increases with the input frequency at the relatively low frequency range (10 − 5,000 Hz), after which the amplitude of SPL approaches saturation. As shown, the FENG-based loudspeaker has a wide coverage of frequency that is sufficient to overlay the auditory perception frequency range of humans that is ∼20 to 20,000 Hz (refs 35,36). Moreover, it can be observed that the larger FENG-based loudspeakers produced larger SPL, and this is expected since larger devices represent larger 'point sources'. Due to the paper-like flexible thin-film structure of the FENG-based loudspeaker, a single-layer device can be folded and stacked to form a multilayer structure, where the giant dipoles in adjacent layers (in contact after folding) have same charge polarity orientation. After stacking, the polarity of each layer (now composed of two FENG surfaces with same polarity) will be opposite to the one below and above, resulting in electric fields pointing to (or leaving from) alternating layers across the device's thickness (see Supplementary Fig. 6). Figure 4b shows the relationship between SPL and the layers of FENG loudspeaker with same area (8 cm × 8 cm). For the stacked multilayer FENG-based loudspeaker, we can see that the SPL increases with the number of layers, similar to the effect we obtained from increasing size of one layer structure (see Supplementary Fig. 7). This indicates that the performance of FENG loudspeaker can be enhanced by increasing layers through simple folding or stacking. The third configuration is designed to obtain the effects of adding a substrate on the frequency response of the FENG. Figure 4c shows the SPL frequency spectrum of a free-standing FENG-based loudspeaker, with and without an acrylic substrate. Although there is a difference between these two SPL amplitude-

frequency responses caused by the substrate, the FENG-based loudspeaker adhered to soft acrylic still produces an SPL of ∼73 dB yellow (a decrease of ∼8 dB in comparison with free-standing configuration) while keeping a similar amplitude-frequency behaviour. To demonstrate the performance and the potential applications of flexible FENG loudspeaker, we fabricated a music-playing Michigan State University flag, as shown in Fig. 4d–f. The developed flag can be waved, rolled or folded—just like any other flag—and has no evident physical or cosmetic difference from any other. A matrix of FENG-based loudspeakers is embedded between two fabric layers. It consists of nine patches of single layer free-standing FENG-based loudspeakers with area of 7 cm × 8 cm. Both sides of their surfaces are electrically connected by copper wires. Like free-standing single layer structure, the matrix of FENG-based loudspeaker is capable of producing sound pressure from both sides (Fig. 3c,d) that easily transmits through the fabric layers. This FENG-based music-playing flag features high-output SPL and wide audio frequency range, as well as lightweight, flexibility that derive from the nature of thin film and the separated matrix configuration. As a demonstration (Fig. 4f and Supplementary Movie 1), a marching band music with a variety of instruments is played by the developed music-playing flag while waving in the air. In addition, the layered thin-film configuration allows the merits of not only being tailored into different shapes and sizes according to applications, but also being placed on a variety of rigid or flexible surfaces, as well as their free-standing use.

**Principle and performance of FENG-based thin-film microphone**. Microphones operate, essentially, in the opposite way than a loudspeaker. Microphones are electroacoustic transducers that transform acoustic vibrations into electrical signals. Today, acoustic sensing has been dominated by electret microphones that have permanently charged dielectric, avoiding the use of DC bias[37–39]. The reversible electromechanical interaction between the electrical and mechanical states of FENG allows the device to operate as a microphone, while still keep its loudspeaker functionality. There are two particularities of FENG that makes its use as a microphone very promising. First, the large charge accumulation on its surfaces give the device a high electromechanical transformation efficiency; this means that large electrical signals can be produced from small mechanical vibrations. Second, the massive giant voids make it a hollow foam film with a low Youn's modulus (1,500 N mm$^{-2}$) (ref. 31) in its thickness direction. Therefore, the external sound pressure generated from music or voice is sufficient to create vibrations along the FENG thickness large enough to produce detectable electrical signals. In other words, sound vibrations from voice or music generate a similar effect on the FENG than applying mechanical pressure perpendicular to the surface. The sensitivity of the FENG-based microphone can be demonstrated by recording the electrical signals produced due to external sound or music, and then comparing the quality of the recorded sound with the original. To this end, a FENG-based microphone system was built to record a famous aria 'La Traviata, Brindisi (Verdi)', as shown in Fig. 5b and Supplementary Movie 2. This testing song is originally performed by a soprano and a tenor, accompanied by symphony orchestra. Thus, different pitches from the two artists and a variety of instruments are included in the test. The tablet's output electrical signal is sent to an audio amplifier that drives a traditional electromagnetic loudspeaker to produce sound wave. The sound wave compresses the nearby air that in turn presses the FENG-based microphone. The generated electric signals from the FENG microphone are then recorded. This setup resembles the typical use of a commercially available microphone,

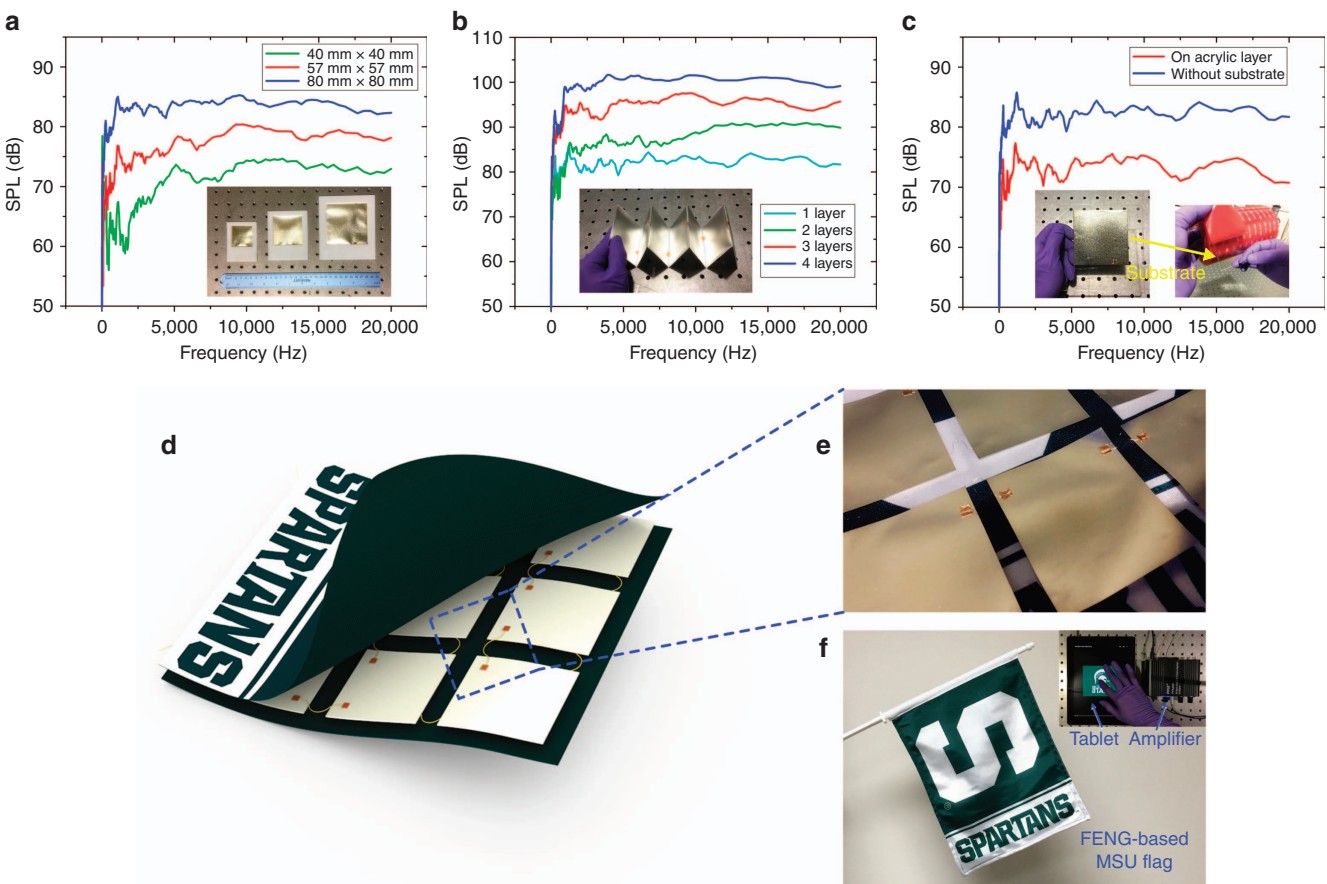

**Figure 4 | Characterization of amplitude-frequency response and FENG-based music-playing flag.** (**a**–**c**) SPL frequency spectrum measurements from 10 Hz to 20 kHz for three types of FENG-based loudspeaker configurations. The insets show optical images of each type of configuration. (**a**) Free-standing configuration with the size ratio of 1:2:4. (**b**) Folded and stacked multilayer configuration. (**c**) With and without substrate configuration. (**d**–**f**) Design and demonstration of Michigan State University (MSU) music-playing flag. (**d**) Schematic diagrams. (**e**) Expanded optical image of the matrix. (**f**) Functionality demonstration (see Supplementary Movie 1).

where an AUX cable is connected to a personal computer with an audio card that can record audio. The direct mechanical-to-electrical energy conversion (Fig. 3b) allows the use of the FENG-based microphone for the real-time recording of the music played by the tablet. The time-domain sound wave signals and the acoustic spectrograms for the original music (tablet's output) and the recorded music (FENG-based microphone's output) are shown in Fig. 5c–f, respectively. It can be seen that the acoustic information of the recorded music is highly similar to that of the original music, even for a small piece. Supplementary Movie 3 demonstrates the FENG-based microphone effectively copies the music or voice it 'hears' with high quality and fidelity. It is worth mentioning that the FENG-based microphone and loudspeaker are essentially the same structure, but employ opposite reversible electromechanical interaction effects, and this means the FENG-based loudspeaker/microphone is genuinely a dual-functional acoustics device. According to the user's requirement, FENG could conveniently switch its functionality.

**FENG-based identity recognition for privacy security application.** Driven by the rapidly increasing demand of global markets, wearable/portable device are continuously pursuing for improvements of human–machine interactions. Along with this trend, wearable/portable flexible electronic devices are poised to collect more and more personal information (for example, social

contact information, daily GPS positioning or even physical activity and healthy status) from users. Therefore, privacy and security will become performance parameters that need to be addressed. The excellent voice recording performance of FENG as well as its light-weight and flexible thin-film structure may open up new possibilities in privacy and security applications of flexible electronics. The most typical security protocol standing between an unauthorized user and a personal computer or a portable/wearable personal gadget with private information is a 'typed password'. In this security protocol, the password text can be either compromised or hacked. The high sensitivity of the FENG-based microphone allows for voice recognition, enabling an extra security layer by replacing the 'password typing' process by a 'password speaking' process. In this security application, the system recognizes both the text password and the authorized user's voice. Given the flexibility and demonstrated wearability of FENG, a FENG device can be discreet and even unnoticeably installed within the wearable gadget or personal computer, and then used to determine the user's identity and grant access. Figure 6a and Supplementary Movie 3 show the process of using FENG as a 'watch-dog' to prevent unauthorized users from accessing to the owner's personal computer even in the case when unauthorized users obtain the correct voice code. Although all testers in this demonstration spoke the correct voice password 'OPEN SESAME', the FENG-based microphone was used to analyse their voiceprints according to designed algorithm and ultimately only give access to the one with voiceprint matching

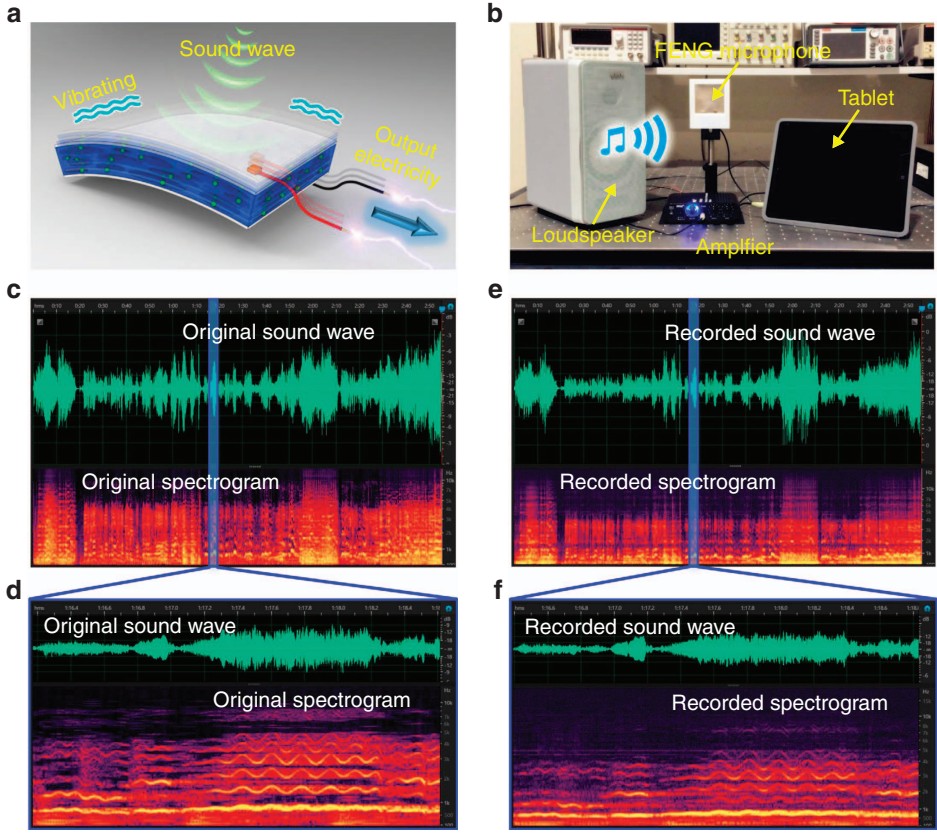

**Figure 5 | Acoustic wave recording performance of FENG-based microphone.** (**a**) Mechanism of the transformation from acoustic energy to electric energy. (**b**) Experiment of FENG-based microphone system for recording music 'La Traviata, Brindisi (Verdi)' (see Supplementary Movie 2). (**c,d**) Sound wave and acoustic spectrogram of (**c**) original music and (**d**) recorded music. (**e,f**) Expanded and detailed view of a small piece of synchronous music for (**e**) original music and (**f**) recorded music.

the previously saved database, which is the authentic user. Herein, we designed Mel-frequency cepstral analysis with artificial neural networks model[40–42] to realize the users' identity recognition based on FENG (Fig. 6b). As the unauthorized user spoke to FENG in request for access, his or her voiceprint information (Fig. 6c–e) was analysed that reflected individual's discrepancy of physical shape of vocal tract[43,44]. The generated voltage due to sound pressure (Fig. 6c) reflects time-domain acoustic information; the spectrogram (Fig. 6d) shows the frequency-domain acoustic information; and the periodogram power spectral density estimate (Fig. 6e) reflects the acoustic energy information. It can be observed that when two testers spoke the same voice code, the FENG-based microphone records their three key voice information, and accurately reveals the user's identity that is indispensable for further privacy protection. Such a simple paper-like, dual-functional acoustic devices will have broad and various promising applications (Fig. 6f). The flexibility of the FENG-based microphone allows for identity recognition system to be integrated not only in a variety of flexible electronic devices, but also portable thin cards, such as ID cards and debit or credit cards. Besides, such advantages might open up new prospects to the design and fabrication of new-style loudspeakers, and the improvement of current active noise cancellations technology for automobiles, aircrafts and submarines and so on.

## Discussion
The applicability of the FENG device is here expanded beyond energy harvesting applications. The strong electromechanical coupling of FENG gives the device the particular dual-functional capability of using sound to produce electricity, and using

electricity to produce sound. The SPL for three different FENG configurations (free-standing, substrate-held and rolled-up FENGs) is characterized as a function of space and frequency in the range of the human audible range. The device is implemented in three systems, demonstrating the application of the device for developing music-playing flexible flags, and flexible patches that can be used for recording music with high fidelity or discreet devices that can reveal the identity of individuals trying to gain access to a personal computer.

## Methods
**Fabrication of FENG-based loudspeaker or microphone.** PP film (fabricated by EMFIT Inc.) containing foreign silicates particles (0.1–10 μm) that serve as stress concentrator was experiences stretching in two perpendicular directions. During this process, high pressure (for example, 5 MPa) nitrogen diffuse into the PP film to make the internal pressure within the voids equal to the external pressure. Then, the external gas pressure was released suddenly that results in the swell of the voids in PP film. To stabilize and stiffen the swelling voids, thermal treatment (for example, > 100 °C) is carried out. Next, a layer of silver (500 nm) was deposited by sputter coating on one surface followed by depositing another layer of silver (500 nm) on the backside after flip. Paper-like configuration allows the film to be cropped into different shapes or sizes with the aid of blade. The final step was connecting two copper wires as electrodes to the silvers layers via copper tape.

**Fabrication of music-playing flag and multilayer structure.** Nine patches of single-layer FENG-based loudspeaker (7 cm × 8 cm) were arranged to form a 3 × 3 matrix. The electrical positive surface and negative surface of all patches are aligned to the same flag's surface end connected to the same electrical node. In other words, all the positive sides of the FENG-based loudspeaker are facing one side of the flag and electrically connected in series, while the negative sides are facing the other side and also electrically connected in series. Copper wires are used for the electrical connections inside the flag. The whole matrix was then embedded into two fabric layers of flag (28 cm × 33 cm). For the multilayer structure, the surfaces with the same polarity of single FENG-based loudspeaker are stacked face to face

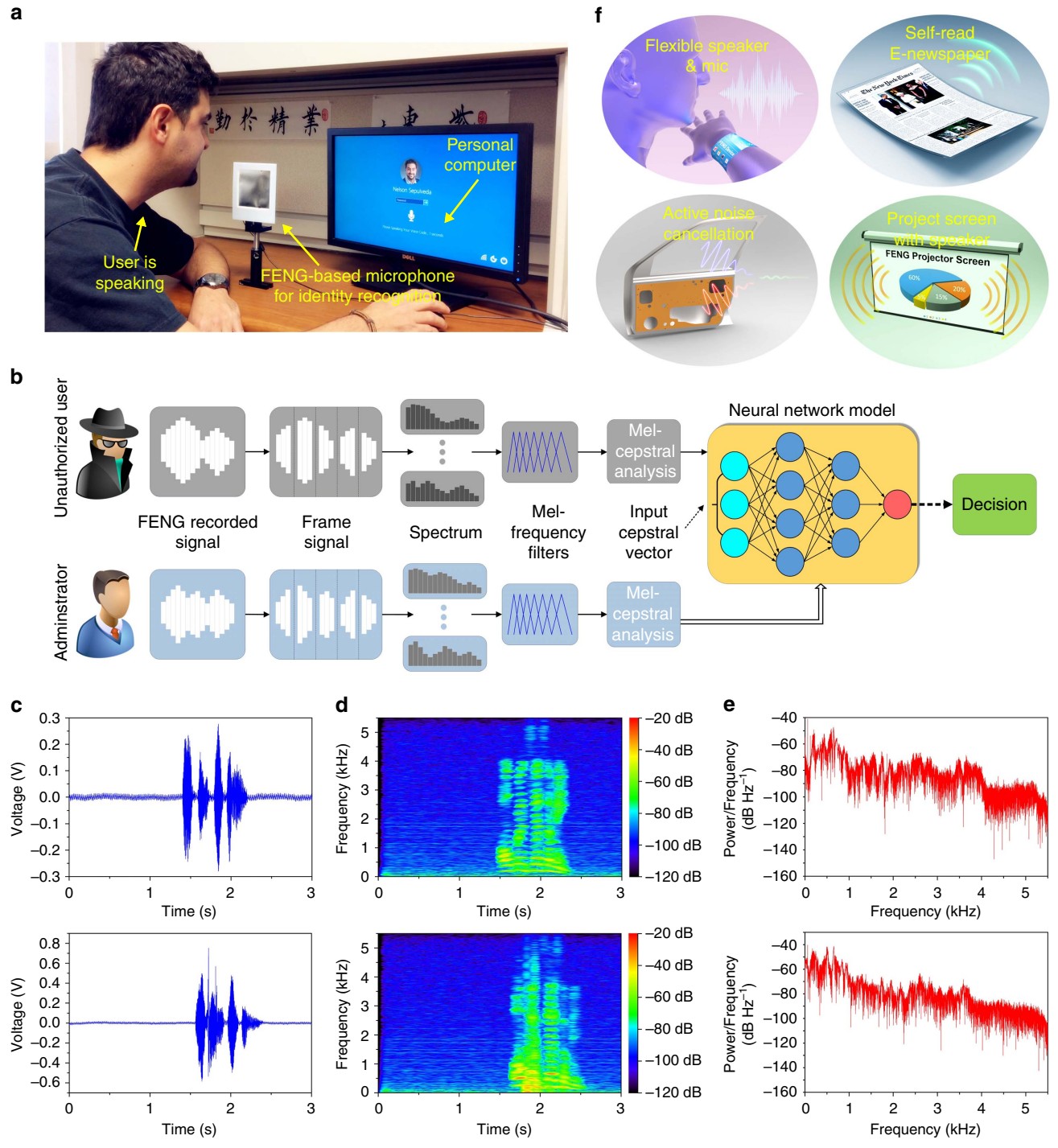

**Figure 6 | Demonstration of FENG-based identity recognition along with other potential applications of FENG. (a)** Requesting access to a personal computer by using speaking voice code to the FENG-based identity recognition system (see Supplementary Movie 3). **(b)** Schematics of recognition algorithms based on Mel-frequency cepstral analysis with artificial neural networks model. **(c–e)** Acoustic information of voice code 'OPEN SESAME' spoken by the administrator (top) and an unauthorized user (bottom). **(c)** Sound wave. **(d)** Spectrogram. **(e)** Periodogram power spectral density estimate. **(f)** Various applications of FENG-based acoustic device.

(that is, in electrical contact). Copper tapes were used to connected adjacent faces. Last, two copper wires were used as positive and negative electrodes.

**Characterization and measurements.** In the SPL directivity measurement, 1/2′ prepolarized Microphone and preamplifier system (378A06, PCB Inc.) with sensitivity of 12.6 mV Pa$^{-1}$ and sensor signal conditioner (484B06, PCB Inc.) were used to measure the sound pressure. Precision potentiometer (Rourns Inc.) read the angle of FENG-based loudspeaker during the rotation process. The input signal is generated by an arbitrary Waveform generator (3,390, Keithley Inc.) through a voltage amplifier (HVA 200, Thorlabs Inc.). Up to 300 and 60 V are used for SPL

directivity and frequency-response measurements, respectively. In the frequency-response measurement, a spectrum/network analyzer (3589A, Hewlett-Packard Inc.) is employed. The distance between the centre of FENG-based device and the head of microphone system for all the measurements is 12 cm. The output signals of all measurement were acquired by an integrated real-time Controller (cRIO-9075) with an analog input module (NI 9201, National Instruments Inc.).

**Data availability.** The data supporting the findings of this study is available from the corresponding author on request.

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

## Acknowledgements

This work was supported by the National Science Foundation (NSF ECCS Award #ECCS-1306311). We are very thankful to Professor Andrew Barnard from Michigan Technological University for his valuable help. We are also appreciative of the Composite Materials and Structures Center (CMSC) at Michigan State University and to Dr P. Askeland for his assistance and suggestions.

## Author contributions

W.L. was in charge of the overall design of the devices, experiments, data gathering and analysis. D.T. and R.D. assisted with testing and experimental setup, respectively. Z.W. and C. Wu contributed with ideas for device design. C. Wang assisted with the design of the devices and the experiments for their testing. Z.L.W. and N.S. assisted in the conception of the initial ideas for the demonstrations, supervision and general guidance for the experiments, analysis of results and preparation of the manuscript. All the authors were included during the discussion of results and participated during the review of the manuscript.

## Additional information

**Competing interests:** The authors declare no competing financial interests.

**Publisher's note**: 

