## [Peer Review File · Nature Communications]

Reviewers' comments:

Reviewer #1 (Remarks to the Author):

The authors present interesting results that will definitely have significant impact on the field of nanogenerators. The present work brings to the table a type of nanogenerator device that can convert energy in both directions: from mechanical to electrical, and from electrical to mechanical domains. They demonstrate the capabilities of the device through three applications: music-playing flag, sound recording thin film patch, and a flexible microphone. This type of dual or bi-directional energy conversion mechanism seems to be unique for the operating principle used in this device. The work is presented very clearly and I am convinced by the significance of the work in the sense of the possibility it is opening. This work demonstrates a flexible patch device that serves dual purpose: energy harvesting AND also a transducer. I am sure that it will be followed by many other researchers in the field.

Therefore, in terms of novelty, impact, and scientific quality, the present manuscript is definitely worth publishing in Nature Communications. However, in the interest of maximizing or optimizing the impact of the communication, the authors should address the following minor concerns.

1. It is understandable that the main contribution of the article is the dual-function or bi-directional operation of the FENG. However, the device has a particularity that is not found in other flexible film-like devices, which is the ability to produce power as well. The authors are too focused on the bi-directional demonstration and are ignoring a major benefit of the device, which is self-powered. In fact, the only part in the paper where self-power is mentioned is in the title and abstract. The authors should look for a way to emphasize (early on in the manuscript, perhaps the introduction) the ability of the FENG device to generate power as well. This part should be highlighted early on to demonstrate the unique impact of the device, since this will be very attractive to the readers.

2. I would have expected to see isotropic directivity for the free-standing configuration. However, Fig. 3d clearly shows directivity along 0, 90, 180, and 270 degrees. Why is this the case? Could the authors run some simulations to understand and perhaps explain this a bit better?

3. Why did the authors focus on the three configurations: free-standing, substrate, and rolled device? There should be some rationale for these three configurations. Does it have something to do with the end-user applications?

4. It would be really nice to see if the authors could define some design guidelines. For example, how is the directivity influenced by stacking? How much is the SPL attenuated by the substrate (for the substrate-held configuration)?

5. There is some confusion about stacking and folding the FENG. Are you expected to obtain the same result from stacking multiple FENG devices than by folding multiple FENGs? How are these two different (if they are actually different)?

6. In Figure 5 c-f some of the figures have added text that is not clear or interferes with important parts of the figure or plot.

Reviewer #2 (Remarks to the Author):

This manuscript demonstrates self-powered loudspeaker/microphone using the ferroelectric nanogenerator (FENG) in various forms. Along with the extensive research on the wearable devices

related with human senses of touch and vision, this acoustic device would have a very high potential in actual applications. The suggested idea is strongly supported with the high level acoustic signals and the all the experiments are performed systematically. So, this reviewer suggests the publication of this paper in Nature Comm as is.

REVIEWERS' COMMENTS:

Reviewer #1 (Remarks to the Author):

This reviewer finds the authors have properly addressed the comments and question previously raised. Therefore acceptance to Nature Communications is recommended.

Reviewer #1:

The authors present interesting results that will definitely have significant impact on the field of nanogenerators. The present work brings to the table a type of nanogenerator device that can convert energy in both directions: from mechanical to electrical, and from electrical to mechanical domains. They demonstrate the capabilities of the device through three applications: music-playing flag, sound recording thin film patch, and a flexible microphone. This type of dual or bi-directional energy conversion mechanism seems to be unique for the operating principle used in this device. The work is presented very clearly and I am convinced by the significance of the work in the sense of the possibility it is opening. This work demonstrates a flexible patch device that serves dual purpose: energy harvesting AND also a transducer. I am sure that it will be followed by many other researchers in the field.

Therefore, in terms of novelty, impact, and scientific quality, the present manuscript is definitely worth publishing in Nature Communications. However, in the interest of maximizing or optimizing the impact of the communication, the authors should address the following minor concerns.

- 1. It is understandable that the main contribution of the article is the dual-function or bi-directional operation of the FENG. However, the device has a particularity that is not found in other flexible film-like devices, which is the ability to produce power as well. The authors are too focused on the bi-directional demonstration and are ignoring a major benefit of the device, which is self-powered. In fact, the only part in the paper where self-power is mentioned is in the title and abstract. The authors should look for a way to emphasize (early on in the manuscript, perhaps the introduction) the ability of the FENG device to generate power as well. This part should be highlighted early on to demonstrate the unique impact of the device, since this will be very attractive to the readers.*

Response:

The authors are glad and humbled by the positive impression of our work. We are also grateful to the reviewer for the suggestion to highlight the “self-powered” feature early in the manuscript. We fully agree with the reviewer that, this information will not only highlight the impact and applicability of the device, but also make the communication more attractive to readers and beneficial to researchers. Following your important suggestion, we emphasize the “self-powered” feature early on in the introduction section (line 35-37).

- 2. I would have expected to see isotropic directivity for the free-standing configuration. However, Fig. 3d clearly shows directivity along 0, 90, 180, and 270 degrees. Why is this the case? Could the authors run some simulations to understand and perhaps explain this a bit better?*

Response:

We would like to thank the reviewer for asking about the directivity shown in Fig. 3d (in the manuscript). As shown in Figure 1 (in this response), the sound pressure level (SPL) for the free-standing configuration is closely related the position of prepolarized microphone system which is used for measurement. With larger projected area, more sound wave radiated into the prepolarized microphone system. Hence, when the relative angle changed from 0 to 360 degrees, the overall SPL captured by prepolarized microphone system experienced an “increase→decrease→increase→decrease” change sequence, which results in the anisotropic directivity of the free-standing configuration. Following your important comments, we’ve further clarified our expressions in this revision (line: 123-124). Besides, we’ve modified the caption of Fig. 3b (in the manuscript). We sincerely hope our explanation in this response letter and improvement to the manuscript clarifies this.

Figure 1. Directivity measurement of the free-standing configuration.

3. *Why did the authors focused on the three configurations: free-standing, substrate, and rolled device? There should be some rationale for these three configurations. Does it have something to do with the end-user applications?*

Response:

Another insightful comment that we truly appreciate. The reviewer is correct. Choosing circular free-standing, with substrate, and rolled cylinder as typical configurations is based on their potential end-user applications. First, free-standing configuration, which is the most basic FENG-based loudspeaker, can produce sound pressure from both surfaces. Therefore, it is capable of radiating sound from not only the front-side, but also the back-side at the same level, which

could allow more sound to be directed to more listeners in practical applications. Second, FENG-based loudspeaker affixed over substrate demonstrates the device's robustness and ease for installation. It shows its potential for being attached onto a variety of materials (i.e. wall, glass, plastic cover, cloth, etc.). Third, rolled cylinder configuration satisfies applications that require omnidirectional sound wave propagation with isotropic SPL.

4. *It would be really nice to see if the authors could define some design guidelines. For example, how is the directivity influenced by stacking? How much is the SPL attenuated by the substrate (for the substrate-held configuration)?*

Response:

The authors fully agree with the comments. Stacking single layer to form multi-layers configuration would increase the overall produced SPL. The SPL directivity would not be able to change because according to our experiments and description in Fig. 3 (in the manuscript), SPL directivity is closely related to the configuration of the device. Stacking would not change the relative position between FENG-based loudspeaker and prepolarized microphone system, nor the projected radiation area during the measurement process. SPL was indeed attenuated by the substrate for the substrate-held configuration, because the substrate blocked sound radiation emitted from the substrate side to a certain extent. Comparing with the SPL obtained from the side without substrate, as shown in Fig. 4c (in the manuscript), a decrease of ~ 8 dB is observed. We've further clarified this in this revision according to your comments (line: 162-163). We sincerely hope our response has made this clear.

5. *There is some confusion about stacking and folding the FENG. Are you expected to obtain the same result from stacking multiple FENG devices than by folding multiple FENGs? How are these two different (if they are actually different)?*

Response:

The reviewer's perception is accurate – stacking multiple FENG devices has the same effect as folding a one-layered device multiple times. During the folding process, the electrodes with the same polarity would be in contact, and the polarization of the giant dipoles alternatively changes in adjacent layers, which is in agreement with the stacked multi-layer device, as shown in Figure 2 (in this response). Consequently, we would say the same result from stacking multiple FENG devices than by folding multiple FENG is expected to obtain due to their virtually identical structures. We sincerely hope our explanation in this response letter clarifies this.

Figure 2. Schematic illustration of the stacked multi-layers structure of FENG-based acoustic device.
(Also see Supplementary Figure 6)

6. In Figure 5 c-f some of the figures have added text that is not clear or interferes with important parts of the figure or plot.

Response:

We appreciate your kindness in pointing out that the added text in Figure 5 c-f interferes with important parts of the plots. Following your constructive suggestions, we've moved the added text so as to avoid this interference.

Finally, thank you so much for your review and valuable advice and suggestions.

Reviewer #2:

This manuscript demonstrates self-powered loudspeaker/microphone using the ferroelectret nanogenerator (FENG) in various forms. Along with the extensive research on the wearable devices related with human senses of touch and vision, this acoustic device would have a very high potential in actual applications. The suggested idea is strongly supported with the high level acoustic signals and the all the experiments are performed systematically. So, this reviewer suggests the publication of this paper in Nature Comm as is.

Response:

We truly appreciate your offering this high praise for our paper. We are glad and humbled by the positive impression of our work, and thank you so much for your review.